# COVID-19 and Its Effects on the Management of the Basic Quality Conditions in Universities of Peru, 2022

**Juana Vargas Bernuy [1], Sam Espinoza Vidaurre [2,*], Norma Velásquez Rodriguez [3], Renza Gambetta Quelopana [4], Ana Martinez Valdivia [4] and Ernesto Leo Rossi [5]**

[1] Faculty of Civil Engineering, Architecture and Geotechnics, Jorge Basadre Grohmann National University, Tacna 23000, Peru; jvargasb@unjbg.edu.pe

[2] Faculty of Engineering, Private University of Tacna, Tacna 23003, Peru

[3] Faculty of Economic and Commercial Sciences, Catholic University Sedes Sapientiae, Lima 15302, Peru; nvelasquez@ucss.edu.pe

[4] Faculty of Architecture and Urbanism, Private University of Tacna, Tacna 23003, Peru; relgambetta@virtual.upt.pe (R.G.Q.); anamartinezv@virtual.upt.pe (A.M.V.)

[5] Newman Graduate School, Tacna 23001, Peru; ernestoalessandro.leo@epnewman.edu.pe

\* Correspondence: samespinozav@upt.pe

**Abstract:** The purpose of this study was to determine the influences of variables, crisis management, distance education, the organizational image, and student satisfaction on the basic quality conditions in Peruvian universities during the COVID-19 pandemic. A quantitative, nonexperimental explanatory approach with stratified random sampling was used. A questionnaire was applied to 513 students from public and private universities in southern Peru who received distance education during the months of September to November 2022. For the analysis of the results, a structural equation model (SEM). A hierarchical linear regression was carried out to test the hypotheses according to the dimensions studied on the student satisfaction scale. The study findings showed that distance education and student satisfaction had positive impacts on the basic quality conditions, while crisis management and the organizational image had a positive relationship but a smaller effect on the quality of the conditions. We concluded that the applied model allows the causal relationships between variables to be explored and that the results will allow university authorities to generate policies that improve their organizational image and crisis management processes and, at the same time, allow them to better plan their crisis management strategies to achieve better satisfaction within the framework of a sustainable university.

**Keywords:** crisis management; distance education; organizational image; student satisfaction; basic quality conditions

## 1. Introduction

COVID-19 has caused various changes around the world, among them those of note in terms of policies and strategies [1] for the development of sustainable education. Added to this are the efforts by the United Nations to achieve progress for all countries. In this sense, the aims are to "promote quality education (SDG 4)", improve the gaps that exist, and generate opportunities to carry out research to determine the conditions of inclusive and equitable quality for all [2]. Furthermore, knowing what has happened in vulnerable areas during this health emergency is essential to guarantee the maintenance of educational quality. Based on the declaration of the World Health Organization (WHO), it is important to determine how remote education has become a basic resource to reduce educational gaps so that educational services are not reduced in quality, especially in places where there is educational heterogeneity and low technological inclusion. As stated in [3], the pandemic has had significant consequences for different areas of life and has also affected the operation of the educational dimension, generating closures and making the

development of face-to-face classes impractical. On the other hand, it has generated the opportunity to transform organizational culture processes in higher education, mainly by venturing into the use of digital technology and improving processes for managerial and political decision-making as well as working with individual and collective actors and interested parties. For their part, the authors of [4] pointed out that COVID-19 has shocked the global health economy, causing policymakers to generate strategies to balance strict public health measures to curb the spread of the virus with the adverse consequences for health, education, and the economy.

In the educational field, the adaptation of new teaching and learning practices during the pandemic has had an effect on students. Higher education generates knowledge and seeks, through high quality teaching, to generate employable graduates who can address the difficulties of the country. These are fundamental parameters that can be used to measure the capacity and quality of a university [5,6]. A comparison of the situation before and after the COVID-19 pandemic not only indicates that higher education is used to develop quality human capital but also highlights the importance of the collaboration of the education sector with others to produce, for example, sustainable industries. The role of technology is crucial to develop better quality study environments. In this regard, online technology has allowed a sustainable path of higher education, whereby teachers at all academic levels have had to adapt their classes to online modalities and use new technologies, despite few having the necessary skills to handle them. Educators have had to modify the way in which exams are administered to cover the content taught and to avoid fraud [7]. In this sense, online teaching and learning is successful when the educational community is involved in meaningful interactions, recognizing that learning can be effective when it is based on discussions between the teacher and students. Thus, interactive learning emphasizes a shift from a high level of control by teachers to more self-centered learning by students. On the other hand, the change from physical to virtual training has become an integral part of the educational system worldwide. However, the levels and methods used to achieve high quality education are varied and depend on the various factors associated with the information and communication technology (ICT) policies and practices used in education [8].

In a world where companies use technology as a development engine, it is crucial to understand the role it plays in educational organizations. In this sense, organizations focus on the quality of services provided as a competitive strategy and image to attract or retain customers. Thus, higher-order educational organizations that generate high quality services are playing an increasingly important role in the development of the economies of many nations [9]. Traditional education was suspended in Peru due to the pandemic, and higher education institutions complied with government directives in light of the spread of the pandemic. This closure lasted longer than expected and continued in the 2020–2022 educational period, in which decisions were made not to interrupt education at universities by adapting their organizational and educational activities without altering their quality. On the other hand, studies in the literature to show how the circumstances of the pandemic affected the basic quality conditions of universities in Peru are lacking, mainly regarding their management in the face of the crisis, the adaptation to remote teaching, and the perceptions of students and effects on the image of the educational institution [1].

Following this introductory statement, we present the research problem and related objectives, followed by the literature review, research context, and design. The results and discussion are then presented, and conclusions are drawn at the end of the paper.

*Research Problem, Objectives and Questions*

The relationship of COVID-19 to, and its effects on, the quality of higher education management is an important topic to study, since its impacts have generated a series of changes and provisions for the proper functioning of higher education in Peru (SUNEDU, 2020) [10]. In March 2020, the Peruvian government generated a series of provisions to prevent a reduction in quality and allow teaching and learning processes to continue despite

the pandemic. In this sense, gaps were identified in terms of education management in public and private universities, crisis management, the incursion into distance education, the role of the organization image, and the quality of the conditions, as well as the impact on student satisfaction of going from face-to-face to remote study, where the control and learning processes depend mainly on the degree of attention of the student. Regarding the control of activities, there was a change from hours of teaching in a classroom to the generation of activities where the "instructions" and communication processes of the teacher are key to the achievement of good learning.

Health emergencies in Third World countries can lead to the collapse not only of the economy but of the economic system. Thus, it is fundamental for studies to recognize that high quality processes that do not diminish the employability of young students can only be generated with the use of competent resources.

Taking into account previous arguments, this research aimed to determine the effects of crisis-management processes, distance-education practices of universities, student satisfaction, and the organizational image on the basic conditions of quality in universities of southern Peru. A representative sample from the south of the country of study (Peru) was used, since the sample was selected without the distinction of regions or geographical locations.

To achieve this objective, the following specific objectives were developed: first, we aimed to understand the effects on the basic quality conditions; second, we analyzed the impact of COVID-19 on student satisfaction and the quality of the conditions; and third, we acknowledged the lessons learned from the effects of COVID-19 on the quality of the conditions. In pursuit of the objectives, the following research questions were posed to deepen the study: How do crisis management, distance education, the organizational image, and student satisfaction have direct effects on the basic quality conditions of universities in the south of the country? To what extent does crisis management have direct effects on the basic quality conditions of universities in the south of the country? In what ways does distance education have direct effects on the basic quality conditions of universities in the south of the country? How does the organizational image have direct effects on the basic quality conditions of universities in the south of the country?

Based on these questions and by following the methodological process, the relationships between the perceptions of university students in the southern zone of Peru regarding crisis management and distance-education practices, their levels of satisfaction, and the image of institution in terms of the basic conditions of quality were determined. Among the main conclusions, we found that the quality of distance education and student satisfaction are key factors that universities in the south must monitor, since they affect the perceptions of the basic quality conditions.

## 2. Theoretical Framework

### 2.1. Higher Education

2.1.1. Sustainable Higher Education

Higher education plays an important role in promoting sustainability, but this can be achieved only when barriers are faced and challenges are overcome [11]. A debate on Education for Sustainable Development (ESD) has been taking place since the Agenda 21 of the United Nations Conference on Environment and Development, 1992. It was addressed at the Second World Summit on Sustainable Development, held in Johannesburg in 2002 and during the United Nations Decade of Education for Sustainable Development, following which the participation of higher education institutions around the world in sustainability increased. Furthermore, the Nagoya Declaration reaffirmed this responsibility in 2014. This allowed the objectives established in Rio de Janeiro to be achieved through the HEIs, supporting the realignment of economic, social, cultural, environmental, and educational objectives. Prior to the Rio +20 Conference, the Higher Education for Sustainability initiative was created, the objective of which was to promote the commitment of higher education

institutions to the teaching and promotion of research on sustainable development, the greening of campuses, and support for local sustainability initiatives.

In 2015, this initiative officially became part of the second priority area of the Global Action Program on Education for Sustainable Development partners' network: "transforming learning and training environments". The same ones are included in SDG 4, in which broader inequalities due to COVID-19 were found in low-income countries, where distance learning remains out of reach for at least 500 million students. In [12], it was shown that not only the graduates but the entire higher education system in which knowledge and skills are generated contribute to the productivity of sectors and the generation of sustainability in economies. Education is the most important factor in the generation of sustainability in a country.

### 2.1.2. Educational Regulations

In the education sector, Peru has certain regulations that contribute to the educational quality. These are covered by the following University Laws [13]: Law No. 30220, which encourages the use of innovation and technology (Framework Law on Science, Technology and Technological Innovation, [14]); Law No. 28303, which considers the performance of the public careers of teachers [15]; and Law No. 30512, which focuses on competitiveness and productivity as well as policies that ensure quality at a higher level. This legal framework is adapted to the needs of universities to maintain the educational quality.

### 2.1.3. Higher Education and COVID-19

Higher education contributes to the improvement of the knowledge and skills of people [16]. In this sense, universities and institutes, through their academic courses and syllabi, impart knowledge that contributes to society through the provision of effective solutions to the multiple problems considered in the SDGs. In the Peruvian case, the horizon of higher education has undergone a large transformation since the creation of the National Superintendence of Higher University Education (SUNEDU), which is responsible for supervising and directing the educational quality. Meanwhile, as a result of the health emergency, the educational ecosystem has had to adapt to virtual study, where the use of technological resources has played an important role in allowing the educational process to continue [17].

Various authors [18–20] have stated that remote education has had recognized successes and trajectories in many countries, given that teacher training processes and investment in technological infrastructure have allowed the academic rhythm to continue. Educational institutions have generated hybrid activities in order to prevent the loss of what they have learned and to comply with training commitments. In itself, the digital competence approach has gained ground in the academic world, allowing greater permissibility in the development of classes through the use of technology, virtual reality, and artificial intelligence [21].

### 2.2. Crisis Management

The crisis response is a reflection of the management capacity of organizations. Various authors [1,22] have studied how the effects of the COVID-19 crisis have threatened institutional objectives, in terms of complying with scheduled activities and generating operational risks, presenting an opportunity to generate measurable prevention mechanisms. In the case of the health emergency, there were signs that marked its beginning. However, at a global level, it was not possible to determine the possible consequences or the most effective way to intervene. Other authors [23–25] stated that crisis-management competencies in universities are related to the attitudes and behaviors of administrators and teachers regarding their knowledge of digitization, rather than the lack of digital infrastructure. Learning allows lessons to be learned and risk prevention to be undertaken to minimize the sociocultural, economic, and political effects caused by the health emergency. These

arguments allow us to consider that crisis management can be studied by considering processes that occur before, during, and after the crisis (Figure 1) [26].

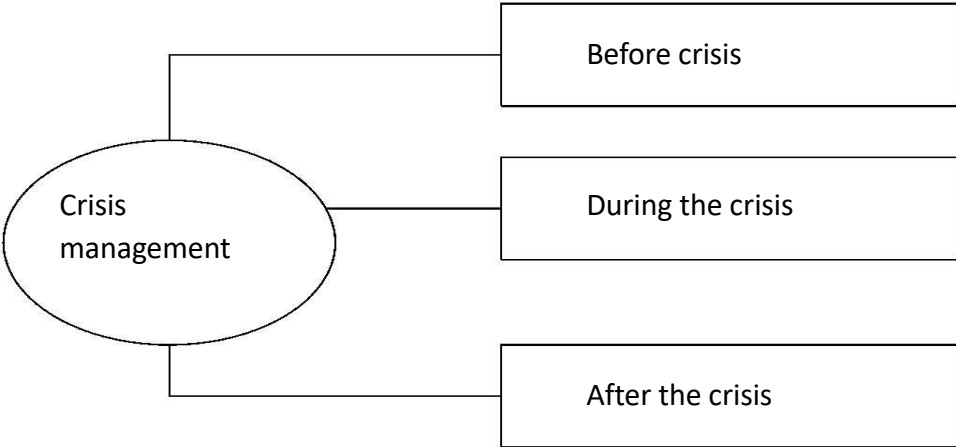

**Figure 1.** Dimensions of crisis management. Note: Instrument designed by Aksu and Deveci (2009) [26].

*2.3. Distance Education*

The technological transformation has generated a different way of transmitting knowledge, creating opportunities for educational organizations. Different authors have highlighted that the evolution of Industry 4.0 has generated changes, for example, in the way in which studies are conducted [27] and the flexibility of time and space for the development of synchronous and asynchronous classes [28]. Regarding synchronous learning, the use of virtual platforms has been essential to connect teachers and students, while asynchronous learning has strengthened interactions through instructions that contribute to knowledge [29]. Other authors have determined the importance of the use of discussion rooms, forums, and video conferences that define some form of "social presence" [30], showing that both types of teaching contribute to the generation of knowledge [31,32]. It has been argued that these methods strengthen continuous learning over time, since educational organizations have improved their offerings of continuous courses that can be learnt at different levels online [27].

Distance education has had direct implications on teaching strategies and didactic approaches. Education continuously evolves to provide society with professionals who can deal with society's problems. Thus, the promotion of distance education and education throughout life has allowed effective methods to facilitate learning to be identified [33]. Furthermore, in a health emergency, this type of education has been the solution to satisfy global educational needs [34], based on the use of technology as a collaborative teaching process and the generation of student autonomy [35]. Other studies have confirmed that the quality of distance education depends on the adequacy of the online infrastructure, qualified personnel, and technological literacy [36].

In this way, universities are designed to develop competencies, build professional skills, and promote the dissemination of knowledge. The dimensions of the distance-education variable were raised by adopting the attitude scale model on distance education offered during the COVID-19 epidemic. These were grouped into five dimensions: satisfaction with the university's distance-education facilities, attitudes towards teachers in relation to distance education, aptitude for online exams, communication and access to distance education, and a comparison of distance education and face-to-face education (Figure 2) [34].

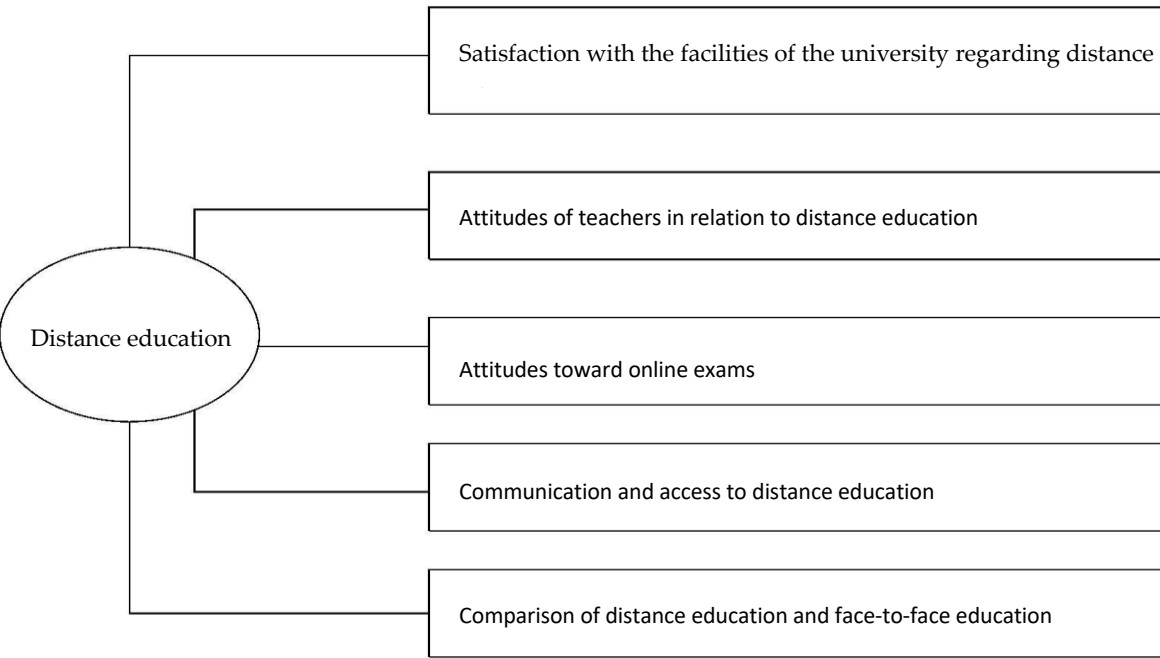

**Figure 2.** Dimensions of distance education. Note: Scale of attitude towards distance education offered during the COVID-19 epidemic (Arslan, 2021) [34].

The paradigm of face-to-face university education in Peru has been abruptly interrupted due to the COVID-19 pandemic and face-to-face, blended, and distance or non-face-to-face teaching have been established. The new paradigm of online education has been introduced in universities for higher education. Universities are obliged to leave their comfort zone and incorporate information and communication technologies for academic development. This is no longer only an academic requirement but also a legal one [37].

*2.4. Organizational Image*

Another important aspect to analyze regarding the quality of service is the organizational image. The reviewed authors define the organizational image as the evaluation carried out by the "client" or "public" regarding the organization. This perception or evaluation is developed through visual, auditory, and behavioral elements. In [38], it was added that, in the context of the educational system, the sum of subjective opinions on the quality of learning and the social environments of various people in the community (internal and external) are a result of their observations and experiences of the institution that accumulate over the long term [39,40].

Authors also identified that, as the internal clients, the students reflect their satisfaction regarding the organization. This type of knowledge generates a sustainable competitive advantage for the university [1]. This information contributes to the ability to choose based on the quality of the institution, and this can be used to assess whether expectations were met during the semester [41]. Therefore, the organizational image positively influences the quality, reputation, prestige, trust, stability, attractiveness, and originality of a university. Thus, the image of a university with good infrastructure, physical resources, social service units that provide socialization, and qualified academic personnel directly affects student success and performance [38,42]. Below in Figure 3 the dimensions of the organizational image variable [42].

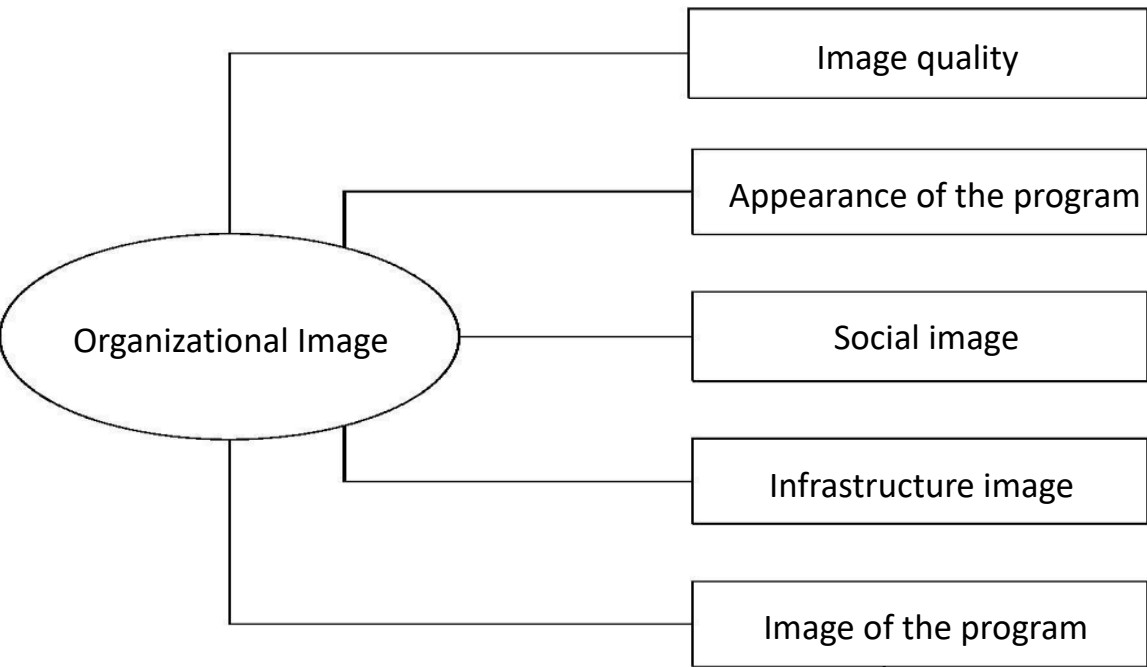

**Figure 3.** Dimensions of the organizational image. Note: Made by Polat, Abat, and Tezyürek (2010) [42].

*2.5. Student Satisfaction*

Student satisfaction is a fundamental variable that is used to recognize the high quality aspects of a university, especially during the pandemic. In this sense, various authors have studied the relationships of student satisfaction in terms of traditional versus virtual learning, which is relevant for understanding how motivation, class design, and tools influence perceptions of educational quality [43,44]. Other authors have studied the process of entering virtuality by students. They found that they were not psychologically prepared for the change and perceived the transition of change as something complex, whereby social interactions maintain demanding academic principles that promote autonomous learning skills favoring student-perceived learning [45].

In addition, a group of studies have developed a measurement of student satisfaction regarding universities. Findings include measurements of the methodology, interface, and adaptability [46], as well as the participation of the students, the utilitarian performance and the perceived quality of the course, and the effectiveness of the contact between teachers and students. These are variables that contribute to the service provided [47]. Below in Figure 4 the dimensions of the Student satisfaction variable [48].

*2.6. Basic Quality Conditions*

Higher education and employability have a fundamental relationship that is associated with the development of the country. Higher education produces qualified human capital according to the standards that are needed in society, since the skills and knowledge acquired in companies or institutions that contribute to their productivity and development are put in place [49].

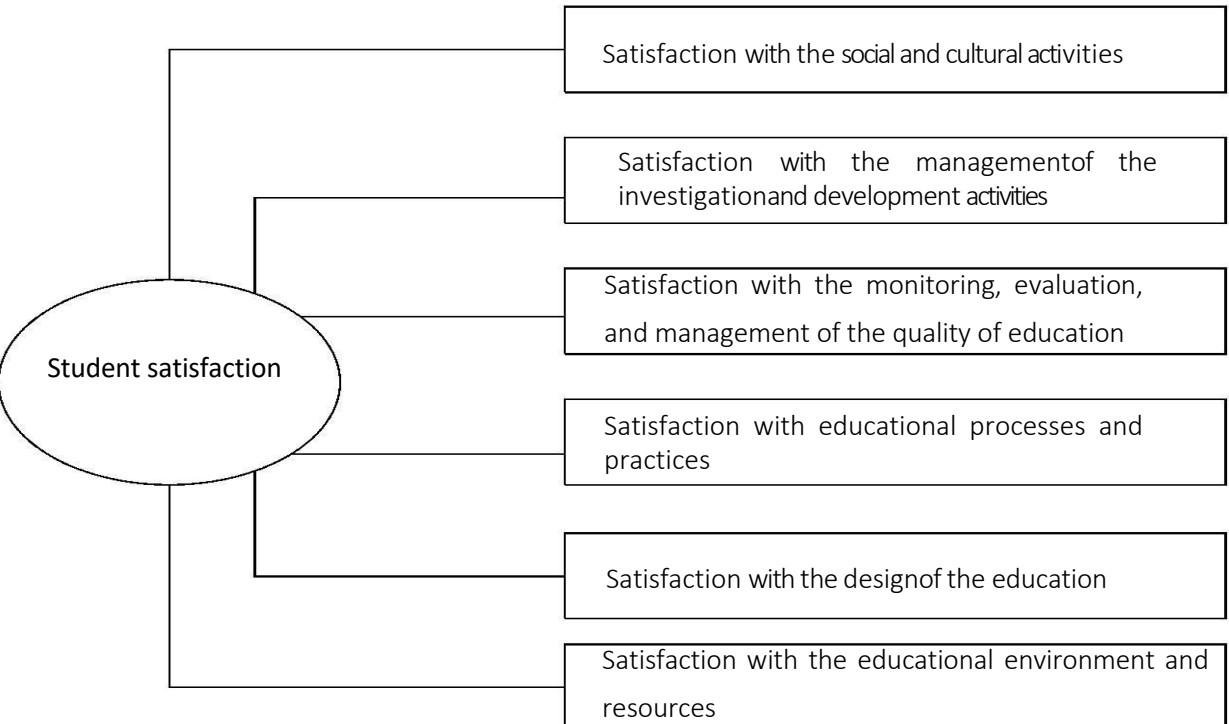

**Figure 4.** Dimensions of student satisfaction. Note: Student satisfaction scale as an indicator in the search for quality in higher education. Produced by Şimşek, İslim, and Öztürk (2019) [48].

Peru is no stranger to this reality, and it seeks to guarantee quality education for all. In this sense, through the university reform, policies have been proposed to regulate actions to ensure educational quality. The National Superintendence of Higher University Education (SUNEDU), in conjunction with the University Law [13], established the Basic Quality Conditions (CBC) for the operation of public and private universities. In article 28, it includes eight basic quality conditions that are linked to the fulfillment of academic objectives: the educational offerings are compatible with the needs of organizations; the required infrastructure and equipment are present; study occurs through lines of research; and there are highly qualified personnel with complementary educational services and labor-insertion mechanisms that can be viewed through a transparent portal and contribute to the decision-making of current and future students. The fulfillment of these basic conditions is associated with university permanence, which leads to changes such as the diversity of the university's educational offerings [50]. Each of the quality conditions is related to teacher performance [51].

SUNEDU, as a result of the health emergency, generated a series of conditions to allow universities to migrate to remote education. Through the resolution of the Board of Directors No. 039-2020-SUNEDU-CD, seven criteria were approved for the supervision of the adaptation of remote education, on an exceptional basis, for subjects taught by universities and postgraduate schools [10]: accessibility, adaptability, quality, availability, follow-up, relevance, and coherence (Appendix A). Non-face-to-face adaptations include the implementation of actions aimed at training with pedagogical tools based on virtual platforms or information and communication technologies which may be necessary for distance learning based on the type of subject. Below in Figure 5 the dimensions of the Basic Conditions of Quality (CBC) [10].

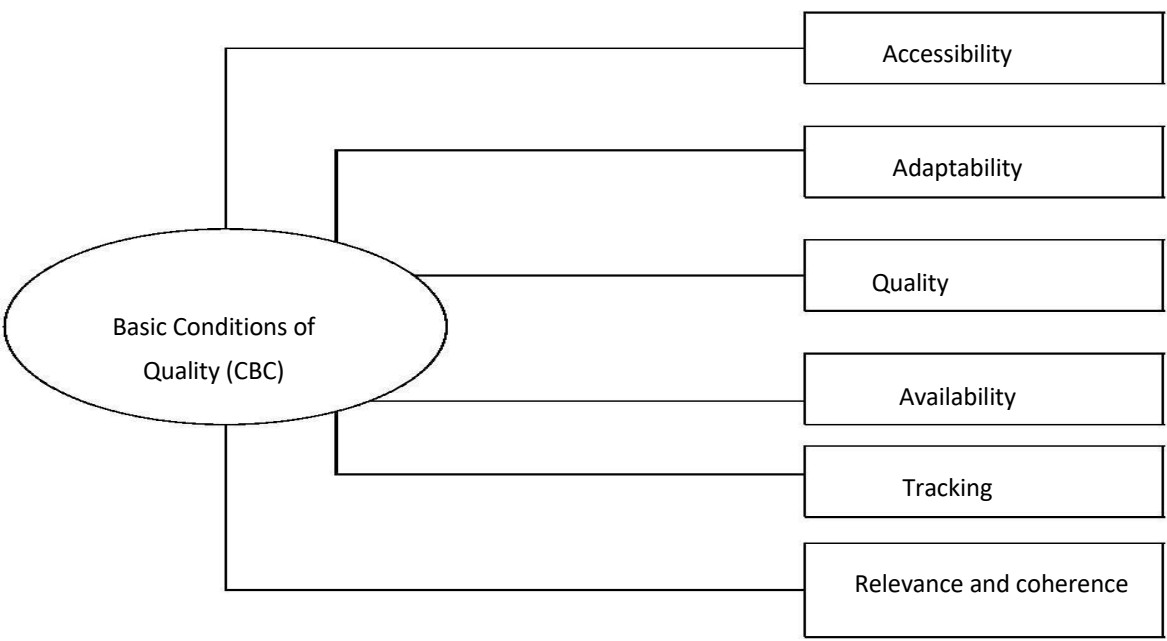

**Figure 5.** Dimensions of the basic quality conditions. Note: Sourced from the National Superintendence of University Higher Education (SUNEDU, 2020) [10].

### 3. Theoretical Model

Each of the mentioned theories helps us to determine the effects of each of these variables on customer satisfaction in a world in which the COVID-19 health emergency has generated unprecedented changes. In Refs. [26,34,42,48], the influences of the following dimensions on the basic quality conditions [CBC] can be identified: crisis management [GC], distance education [ED], organizational image [IO], and student satisfaction [DE]. These dimensions are shown in the structural model. Figure 6 [26,34,42,48] shows the direct and indirect effects associated with the relationships of the variables.

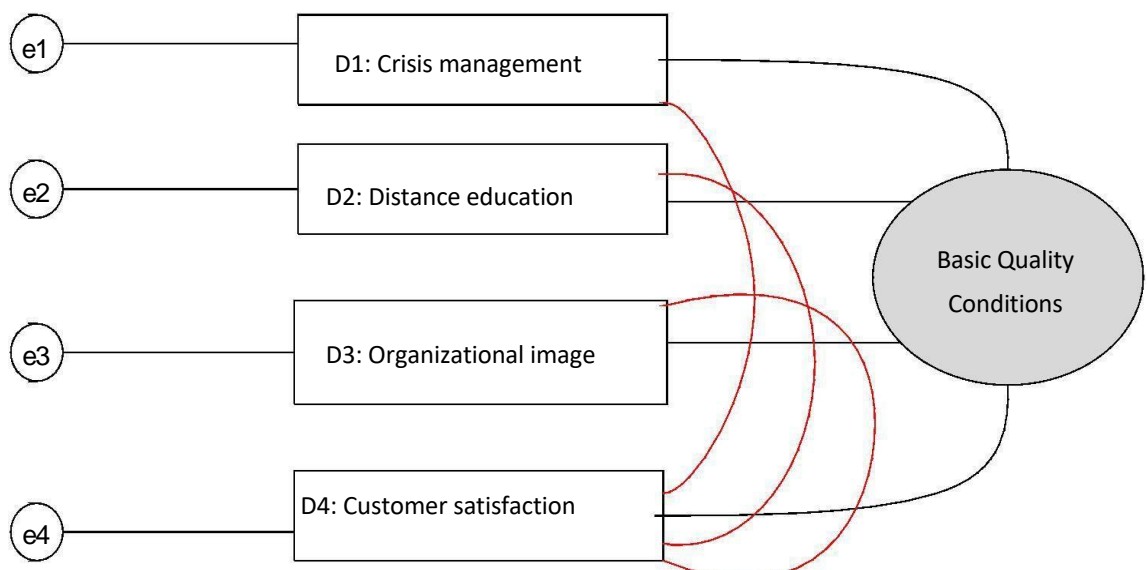

**Figure 6.** Dimensions of the basic quality conditions. Note: Based on [26,34,42,48]—direct effect–indirect effect.

As seen in the bibliographic review, there is theoretical and empirical evidence for the roles of each of these relationships in the field of educational administration. In [52],

the measurement of the degree of satisfaction to which the person (student) is satisfied/disappointed with the form of organization of the institution is shown to be in accordance with their expectations. Therefore, this study sought to first measure how dimensions that have been altered by the health emergency first affected the customer (in terms of satisfaction) and, in turn, were reflected in the basic quality conditions. The other dimensions also substantially impacted CBCs.

*Hypotheses*

1. Crisis management, distance education, the organizational image, and student satisfaction have direct effects on the basic quality conditions of universities in the south of the country.
2. Crisis management has direct effects on the basic quality conditions of universities in the south of the country.
3. Distance education has direct effects on the basic quality conditions of universities in the south of the country.
4. The organizational image has direct effects on the basic quality conditions of universities in the south of the country.
5. Student satisfaction has direct effects on the basic quality conditions of universities in the south of the country.

## 4. Data and Methodology

The field work for the data collection was carried out in the third quarter of the year 2022, in the fourth wave of the COVID-19 pandemic. For this reason, a cross-sectional explanatory design was used. In the Peruvian case, there is a lack of a temporal dimension related to the subject of study, and therefore the answers refer to the moment of data collection and the data evaluation process is considered to represent the period of September 2022. The descriptive methodology related to the model was used to examine the relationships between the crisis management of the universities in southern Peru, distance education, the image organizational structure, and student satisfaction with the basic quality conditions proposed by SUNEDU to deal with the health emergency.

### 4.1. Sampling

The study sample consisted of students from eight public and private universities in southern Peru. The reason for studying the southern zone of Peru is that there are 76,266 students in higher education living under different social conditions. A total of 513 questionnaires were considered valid for data analysis, more than suggested by the stratified random sampling with a proportional allocation carried out (298 samples). The data were collected from students in Arequipa (Catholic University of Santa María, La Salle University, National University of San Agustín), Tacna (Jorge Basadre National University, Private University of Tacna), Puno (National University of Juliaca, National University of the Altiplano), and Moquegua (National University of Moquegua).

For the field process, the questionnaire was sent in electronic format to the department heads of each university so that they could provide the instrument to participants who wanted to participate online. The online form was sent to all students. Participation was completely voluntary, and no information that could reveal the identity of the students was used; rather, questions were asked to demographically characterize the universities. The questionnaire remained open throughout the third quarter, and access to the form was closed at the end of the period.

To gain a better understanding of the study data, the sample size, descriptions of the universities, and the frequencies of the demographic factors of the participants are presented in detail in Table 1.

**Table 1.** Sociodemographic characteristics of the students, 2022.

| Variables | Frequency | Percentage |
|---|---|---|
| **Total** | **513** | **100.0** |
| **Sex** | | |
| *Men* | 293 | 57.2 |
| *Women* | 220 | 42.8 |
| **Age (years)** | | |
| *Average* | 22 | |
| *Median* | 21 | |
| *Mode* | 20 | |
| *Minimum* | 17 | |
| *Maximum* | 48 | |
| **Age range** | | |
| *Under 21 years old* | 290 | 56.6 |
| *From 21 to 25 years* | 163 | 31.7 |
| *Over 25 years* | 60 | 11.7 |
| **City** | | |
| *Arequipa* | 277 | 54.1 |
| *Moquegua* | 22 | 4.3 |
| *Fist* | 80 | 15.6 |
| *Tacna* | 134 | 26.1 |
| **University** | | |
| *Catholic University of Santa Maria* | 101 | 19.7 |
| *La salle university* | 7 | 1.4 |
| *National University of San Agustin* | 176 | 34.3 |
| *Jorge Basadre National University* | 89 | 17.3 |
| *Private University of Tacna* | 47 | 9.2 |
| *National University of the Altiplano* | 67 | 13.1 |
| *National University of Juliaca* | 12 | 23 |
| *National University of Moquegua* | 14 | 2.7 |

Note: Own elaboration based on the applied questionnaires.

*4.2. Study Tools*

The measurement tools were created by combining the dimensions from the correlations and forming a hierarchical linear model in order to identify the determinants of the basic quality conditions (CBC) within each of the dimensions considered in the pandemic stage. For this, the perceptions of student satisfaction with the modality of distance education, crisis management, and the organizational image were considered. The details of each dimension are summarized below.

1.  Crisis management dimension [GC]: this dimension was built with the information on the competencies that emerged for the universities in the south of the country during the pandemic. In other words, we sought to identify the perceptions of the signs of crisis at the universities before the pandemic. This included the will to generate strategies and scenarios related to the educational quality and training as well as the detection of the risks that the crisis could cause. This dimension contains seven items, and the scale developed by [26] was used. Perceptions before the crisis [$\alpha = 0.773$], during the crisis [$\alpha = 0.789$], and after the crisis [$\alpha = 0.819$] were assessed. A global scale value of Cronbach's alpha of [$=0.902$] was obtained.

2.  Distance-education Dimension [ED]: this covered the opportunities offered by universities in the south of the country to students during the period of the health emergency. The measurement scale used was that of [34], which consists of 21 items with five subdimensions to investigate the competencies of teachers regarding distance education; students' attitudes towards online assessments; a comparison with face-to-face education; and communication processes and access to distance education. Satisfaction with the opportunities offered by distance universities [$\alpha = 0.565$], attitudes towards professors in distance education [$\alpha = 0.624$], attitudes towards online exams

[α = 0.474], communication and access in distance education [α = 0.592], and the comparison with face-to-face education [α = 0.499] were assessed.

3.  Organizational Image Dimension [IO]: this dimension was used to analyze the image of universities in the pandemic period. The purpose of the dimension was to determine the perceptions of the students regarding the quality of their academic lives and the relationship with the world of work. The items considered were in accordance with [42]. The Cronbach's alpha value for the global scale was [=0.877].
4.  Student Satisfaction Dimension [SE]: this scale includes 45 items and had the purpose of obtaining information on the general perceptions of students about their universities. The Simsek, Islim and Öztürk scale [48] was used. It consists of six dimensions. The Cronbach's alpha value for the global scale was [=0.896].
5.  The Basic Quality Conditions [CBC]: this dimension contains six subdimensions and is used to measure the basic quality conditions based on the SUNEDU recommendations [10]. The value of Cronbach's alpha for the global scale is [=0.879].

### 4.3. Data Analysis

The study data were analyzed using the statistical package SPSS 26, IBM Amos 22 software, and Stata. The analysis began with descriptions of the dimensions with their respective scales. Following this, correlations were identified and the econometric model was applied. The model was tested with respect to the given variables.

### 4.4. Findings and Interpretation

Findings of the Dimension Analysis

The product correlation analysis was carried out using the Pearson coefficient to determine the relationships between the independent and dependent variables and thus to evaluate crisis management, organizational image, distance education, student satisfaction, and the basic quality conditions.

Table 2 shows that the reliability of the scales is well above the limit, so they are acceptable [α = 0.70]. The descriptive statistics indicate that the lowest average value occurred for distance education [2.4951] and the highest average value was obtained for the organizational image [2.6296] and student satisfaction [2.6043]. Regarding the relationship coefficients between the continuous (independent) explanatory variables, positive relationships of varied levels were detected between the variables [0.478/0.718]. In this sense, the management of the crisis in universities, the organizations' image, and the levels of student satisfaction during COVID-19 increased positively, affecting the basic quality conditions in the same way. To check that there were no multicollinearity problems, the VIF (variance inflation factor) values were considered. Since the values were all lower than the critical value of 10, which corresponds to the maximum accepted value, it is concluded that there were no multicollinearity difficulties [53].

**Table 2.** Descriptive statistics and Pearson relationships between variables, 2022.

| Dimensions | Mean | SD | α | VIF | GC | DE | IO | HE | CBC |
|---|---|---|---|---|---|---|---|---|---|
| **GC** | 2.5419 | 0.56484 | 0.902 | 1804 | 1 | 0.575 ** | 0.510 ** | 0.532 ** | 0.519 ** |
| **DE** | 2.4951 | 0.51961 | 0.899 | 1687 | | 1 | 0.519 ** | 0.478 ** | 0.522 ** |
| **IO** | 2.6296 | 0.52222 | 0.877 | 2371 | | | 1 | 0.718 ** | 0.616 ** |
| **HE** | 2.6043 | 0.54606 | 0.896 | 2276 | | | | 1 | 0.760 ** |
| **CBC** | 2.5750 | 0.55439 | 0.879 | | | | | | 1 |

Note: N = 513; SD: standard deviation; α: Cronbach's alpha; VIF: variance inflation factor. ** The correlation is significant at the 0.01 level (bilateral).

The Pearson correlations between all variables were below 0.80 (Figure 7) A structural model (SEM) was developed, and an analysis of the equation model was carried out to determine the relationships between the variables.

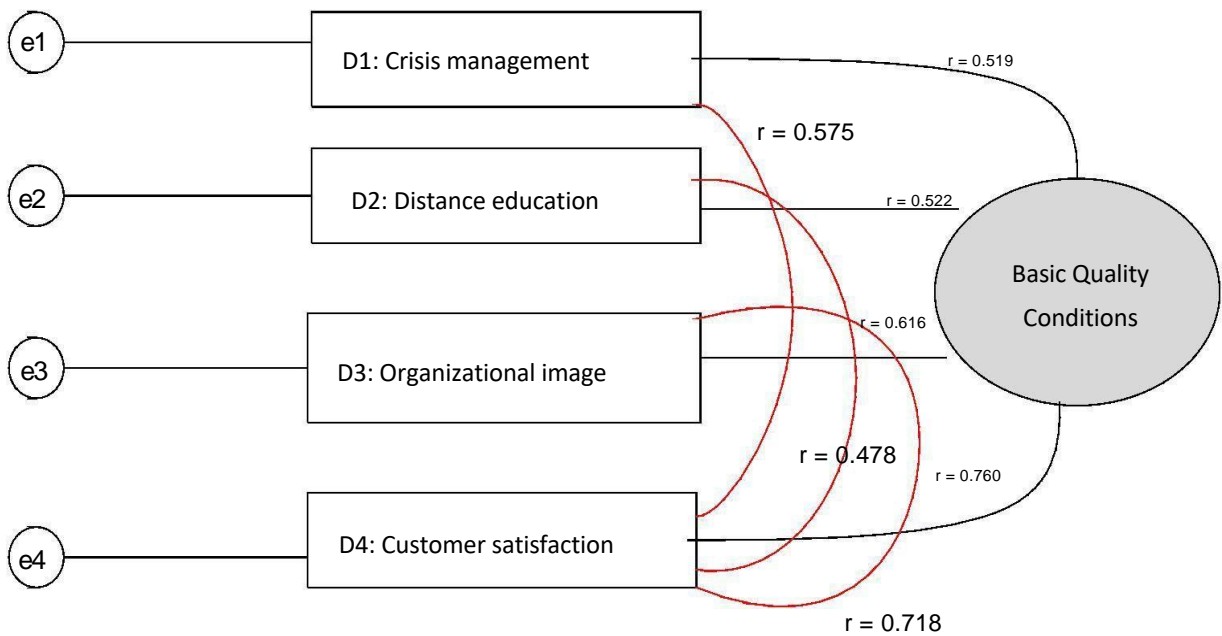

**Figure 7.** Dimensions of the basic quality conditions and their correlations. Note: N = 513.

In order to identify the relationships between the variables—crisis management, distance education, organizational image, customer satisfaction, and the basic quality conditions—a multivariate analysis was carried out. The nature of our dependent variable (continuous variable that follows a normal distribution) justifies the use of a linear regression. Table 3 presents the results of the hierarchical linear regression that included the variables of interest in stages.

**Table 3.** Linear hierarchical regression. Dependent variable: basic quality conditions (N = 513).

|  | **MODEL 1** |  | **MODEL 2** |  |
|---|---|---|---|---|
| *Control variables* |  |  |  |  |
| Type of university | −0.071 |  | 0.011 |  |
| City | −0.184 |  | 0.051 |  |
| University | −0.110 |  | −0.044 |  |
| Age | −0.074 |  | −0.011 |  |
| *Independent variables* |  |  |  |  |
| Crisis management |  |  | 0.075 | ** |
| Long-distance education |  |  | 0.157 | *** |
| Organizational image |  |  | 0.079 | ** |
| Student satisfaction |  |  | 0.593 | *** |
| Constant | 0.115 | *** | 0.140 |  |
| R2 | 0.087 |  |  |  |
| Increase in R2 |  |  | 0.530 | *** |

Note: ** $p < 0.05$, *** $p < 0.01$.

The base model corresponds to Model 1. This model includes the following control variables: the type of university, city, university, and age. It presents an R2 value of 0.087, which indicates that this model explains 8.7% of the variance in the dependent variable. No significant relationships between the control variables introduced and the basic quality conditions were observed.

Model 2 includes, along with the control variables introduced in the base model, the independent variables considered for the study: crisis management, distance education, the organizational image, and student satisfaction. In this model, R2 increased to explain 53% of the variance, which represents the greater explanatory power of the model on the basic quality conditions. Specifically, the increase in R2 with respect to the previous model

was 44.3%, thus showing an improved goodness-of-fit. The results indicate significant relationships between the dimensions and the basic quality conditions. The results suggest a significant and positive relationship between the dimensions and the basic quality conditions. As seen in the theoretical section, the results confirm that the students considered that there is a relationship between the form of management of the university in a pandemic and the basic quality conditions. The satisfaction of students and long distance education are the factors that had the greatest impacts on the basic quality conditions.

*4.5. Discussion*

The findings of the study are valuable, since they highlight that the activities carried out by both public and private universities in the south of the country have minimized risks due to the effects of the pandemic, a result that is related to the direct and indirect effects found in [1].

This study sought to delve into the individual backgrounds of each of the dimensions studied to validate the emergency regulations implemented by SUNEDU in health emergencies. This contributed to the recognition that, at the national level, there has been a lack of studies that have deepened educational management in health crises.

Specifically, the determinants of four dimensions (crisis management, distance education, organizational image, and student satisfaction) related to the basic quality conditions were explored. Studies conducted in Europe have recognized the importance of investigating the characteristics of each of these dimensions to identify how crises generate direct and indirect impacts on educational organizations according to senior management decisions. The senior management needs to evaluate the advantages and disadvantages of decisions made around education, since these final decisions are evaluated by the governing body and have impacts as suggested by the results obtained.

With respect to previous studies on distance education, it is interesting to observe what has happened in other situations, especially in the study unit, are that there have been significant differences among the decisions made by public and private entities, which have usually been coupled with uncertain situations. For this reason, as confirmed by the results obtained on the perceptions of students, the decisions made by their universities in the pandemic have had positive effects. The positive effects confirmed by the results obtained indicate how important it is for organizations to continue to generate activities at a distance.

The results on the distance-education hypotheses and the direct effects on the basic quality conditions of universities are in line with those of [30,34,54], who affirmed that there is a positive and significant relationship with the basic quality conditions of universities.

Another result that is similar to [38,55] is that students considered the organizational image to be relevant and to influence the basic quality conditions, since it has a strong impact on the employability of students. This result suggests that, for educational managers, it is important to take advantage of the new job opportunities of young people. Therefore, a prospective analysis allows their students to face new challenges and develop their full creative potential so that they can overcome different crises, both internal and external to the organization.

Additionally, in terms of student satisfaction, there is a positive relationship, which can be compared to the results of [6] who, in their comparative study, showed that students had poor academic performances before the pandemic compared to their counterparts during the pandemic. On the other hand, pre-pandemic graduates achieved better job readiness scores, including aptitude and practice scores. These are fundamental characteristics that contribute to satisfaction.

## 5. Conclusions

It is essential for universities to generate sustainable policies that maintain the basic quality conditions. The nature of relationships and the form of organization are factors that contribute to the creation of conditions associated with better student satisfaction. Long-

distance education generates new opportunities, as it can make top-ranking universities in the south of Peru accessible to students not only from the south, but worldwide, leading to new internal and external students and job opportunities and improving processes of international exchange, as established in the framework of sustainable education.

On the other hand, having a good organizational image and monitoring the opinions of students through satisfaction surveys is crucial to generate not only a commitment to continuity but also opportunities for change in universities. Furthermore, having committed and trained teaching and managerial administrative staff is key to generating competitiveness in educational institutions. If possible, unique policies that are adjusted to improve teaching standards and generate interdisciplinary activities within and outside the organization should be implemented. Special interest should be given to the results for student satisfaction and distance education, as they are the factors that influence the basic quality conditions to the greatest extent and, therefore, they are the factors on which policies related to management within the houses of study should be based. Students who are more satisfied with the level of education are more likely to generate better employability ties, promoting greater survival of the organization.

The evaluated literature shows that the COVID-19 pandemic has generated considerable changes, not only from the perspective of educational policies, but also in terms of the capacity of human resources to develop the motivation and distance training of students. The practices learned in the pandemic stage, such as the automation of procedures and the use of virtual platforms, should not be lost because they contribute to the continuity of the educational institution, generating a cost–benefit balance for the organization.

This study has some limitations. First, the data were collected at a single moment of time, which makes it impossible to establish relationships and measure the sequential impacts of decisions made in the educational entities of the south. Future longitudinal studies in each of the universities following the applied questionnaire could allow the better management of crises and overcome this limitation. Second, the application of qualitative tools such as in-depth interviews could improve the results obtained, allowing a more profound understanding of the study unit. Future lines of research and studies could be generated through the applied questionnaire, thus also incorporating other variables that contribute to better organizational decisions. Finally, new studies could also try to deepen the knowledge on the relationship between student satisfaction and distance education through the use of probabilistic models that allow better comparisons in educational organizations.

**Author Contributions:** Conceptualization, R.G.Q. and J.V.B.; methodology, S.E.V. and N.V.R.; software, E.L.R. and N.V.R.; validation, S.E.V., R.G.Q. and A.M.V.; formal analysis, N.V.R.; investigation, J.V.B.; data curation, N.V.R.; writing—original draft preparation, S.E.V. and R.G.Q.; writing—review and editing, J.V.B., S.E.V.; visualization, A.M.V., R.G.Q.; supervision, N.V.R.; project administration, J.V.B. All authors have read and agreed to the published version of the manuscript.

**Funding:** This research was financed by the National University Jorge Basadre Grohmann—Tacna, through the Research Project: "COVID-19 and its effects on the management of basic quality conditions in the universities of the southern macro region of Peru in the years 2019–2021", approved with Rectoral Resolution No. 10097-2022-UNJBG.

**Institutional Review Board Statement:** Does not apply, the study does not require ethical approval, at the level of the institution that finances the research we abide by the Regulation of Originality and Similarity of Research Works and Intellectual Production of the UNJBG and with the Code of Ethics for research of the University National Jorge Basadre Grohmann.

**Informed Consent Statement:** Informed consent was obtained from all the subjects involved in the study in the application of the surveys, which may be shared with the journal through a drive.

**Data Availability Statement:** The database of statistics and the database of the syntax of the software applied for the publication, being their own data generated by the researchers, which can be shared with the journal through a drive.

**Conflicts of Interest:** The authors declare that they have no conflict of interest.

**Appendix A**

| Criterion | Definition |
|---|---|
| *Accessibility* | Non-face-to-face adaptation seeks to facilitate accessible learning alternatives, materially and economically, for students. |
| *Adaptability* | Non-face-to-face adaptations are oriented to the type of subject, its corresponding activities, and the instruments used to measure the achievements of the students. Educational strategies are adapted to non-face-to-face teaching, in line with the COVID-19 prevention and control measures. |
| *Quality* | Non-face-to-face adaptations seek to provide high quality conditions similar to those of face-to-face provision, taking their unique features into account. |
| *Availability* | The implementation of non-face-to-face adaptations ensures the timely provision of services and their availability without unjustified interruptions. |
| *Tracking* | Timely monitoring of changes in the academic planning of subjects and their respective development is ensured. |
| *Relevance and coherence* | The academic departments, postgraduate units, or bodies in which they take place must ensure the coherence and relevance of the non-face-to-face adaptations, according to the content of each academic program. |

Note: National Superintendence of University Higher Education (SUNEDU, 2020) [10].

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
