# Peer review of "COVID-19 and Its Effects on the Management of the Basic Quality Conditions in Universities of Peru, 2022"

_sustainability, doi:10.3390/su15086523_

Round 1

Reviewer 1 Report

COVID-19 and its Effects on the Management of Basic Quality Conditions in the Universities of Peru, 2022 - this paper has discussed a relevant issue and it offers some interesting insights. Yet the paper reaches to the publishable standard. Let me have some comments to be noted fir each part to improve further.

Abstract has mentioned the purpose of this study which is very straightforward – yet the importance is not highlighted. Information on research method / deign is not articulated; rather a sample number 513 is stated without justification. The implication is yet to noted. Hence the abstract demands a rewriting.

Introduction:

A good attempt is made to related with UN agenda. However, the problem of this research is not focused to a specified agenda one. I would suggest that the author would develop a sub-topic so that the problem, scope, gap, research aim and objective along with research questions come under a solid structure. Having said that the research needs to explore the research gap on sustainable education and education in the era of Covid-19. I would suggest that the author should revisit these papers and use them-

# The COVID-19 Pandemic as an Opportunity to Foster the Sustainable Development of Teaching in Higher Education- https://doi.org/10.3390/su12208525

# Online technology: Sustainable higher education or diploma disease for emerging society during emergency—comparison between pre and during COVID-19- https://doi.org/10.1016/j.techfore.2021.121034

The debate and definition on sustainable education is also needs to be explore hence, the author should look into these papers and use them to generate a link either with sustainable education or sustainability in education.  

# Conceptualization of sustainable higher education institutions, roles, barriers, and challenges for sustainability: An exploratory study in Portugal- https://doi.org/10.1016/j.jclepro.2016.11.010

# Sustainable Education and Sustainability in Education: The Reality in the Era of Internationalisation and Commodification in Education—Is Higher Education Different?­ DOI: 10.3390/su15021315

In the literature review the norms of higher education and its role are not explored. Before talking about higher education management and to explore this issue, this needs to be developed. Please look following papers and generate the debate in this area:

# Defining the essence of a university: lessons from higher education branding- https://link.springer.com/article/10.1007/s10734-008-9155-z

# What makes a difference for further advancement of engineers: socioeconomic background or education programs?- DOI: 10.1007/s10734-021-00741-4

# Does GATS’ Influence on Private University Sector’s Growth Ensure ESD or Develop City ‘Sustainability Crisis’—Policy Framework to Respond COP21- DOI: 10.3390/su13084520

Research deign: why quantitative method is used is not justified. The justification of sample is not clear enough – why such sample is made and the basis of math that is used to justify the sample. More information and justification are needed for instrument development and its reliability and validity.

Results are somehow interesting but they need to be presented thematically or coherently either to answer the research question or to prove the hypothesis. However, an extensive discussion is to be made by the back of the literature.

Both theoretical and practical implications are to be noted before moving to the conclusion.

Technical issue-

# The sub-topic is too long and very confusing.

# Language needs a serious editing.

# literature is not very relevant    

This paper needs an extensive revision to reach in a publishable standard      

Author Response

  Dear reviewer, We submit the changes to the article according to your suggestion, the article, and the translation certificate. Sincerely, the research team.   Respuesta al revisor 1 Comentarios
(x) Se requiere una edición exhaustiva del idioma y el estilo en inglés.
Se utilizó el traductor de MDPI Language Editing Services.
Punto 1:
El resumen ha mencionado el propósito de este estudio, que es muy sencillo, pero no se destaca la importancia. La información sobre el método/diseño de la investigación no está articulada; más bien se indica un número de muestra 513 sin justificación. La implicación aún no se ha observado. Por lo tanto, lo abstracto exige una reescritura.
Respuesta 1: Proporcione su respuesta para el punto 1.
Se ha reescrito el resumen, incorporando la importancia del tema, así como el enfoque metodológico y el tipo de muestreo utilizado. Se adjunta el artículo de actualización del resumen.
Punto 2:
Introducción:
Se hace un buen intento de relacionarse con la agenda de la ONU. Sin embargo, el problema de esta investigación no está enfocado a una agenda específica. Sugeriría que el autor desarrolle un subtema para que el problema, el alcance, la brecha, el objetivo de la investigación y el objetivo, junto con las preguntas de la investigación, formen parte de una estructura sólida. Dicho esto, la investigación debe explorar la brecha de investigación sobre educación sostenible y educación en la era de Covid-19. Sugeriría que el autor revise estos artículos y los use-
Respuesta 2: Proporcione su respuesta para el punto 2.
Se ha incorporado la bibliografía sugerida, así como los subtítulos para una mejor clarificación del tema, incorporando problemas, objetivos y preguntas problema. Se adjunta el artículo actualizado.
Punto 3:
En la revisión de la literatura no se exploran las normas de la educación superior y su papel. Antes de hablar sobre la gestión de la educación superior y explorar este tema, es necesario desarrollarlo. Consulte los siguientes trabajos y genere el debate en esta área:
Respuesta 3: Proporcione su respuesta para el punto 3.
Se ha incorporado la bibliografía sugerida y las definiciones de educación sostenible, como se sugiere, una sección de la normativa vigente sobre educación superior peruana. se ha incorporado la gestión. Se adjunta el artículo que actualiza lo indicado.
Traducido del español al inglés - www.onlinedoctranslator.com
2
punto 4
Diseño de la investigación: no se justifica por qué se utiliza el método cuantitativo. La justificación de la muestra no es lo suficientemente clara: por qué se hace esa muestra y la base matemática que se usa para justificar la muestra. Se necesita más información y justificación para el desarrollo del instrumento y su confiabilidad y validez.
Respuesta 4: Proporcione su respuesta para el Punto 4.
Este punto fue actualizado en la sección 4.2. Herramientas de estudio
Las herramientas de medición se crearon combinando las dimensiones de las correlaciones y formando un modelo lineal jerárquico para identificar los determinantes de las condiciones básicas de calidad (CBC) dentro de cada una de las dimensiones consideradas en la etapa de pandemia. Para ello, se consideraron las percepciones de satisfacción de los estudiantes con la modalidad de educación a distancia, el manejo de crisis y la imagen organizacional. Los detalles de cada dimensión se resumen a continuación. Se adjunta el artículo actualizado.
punto 5
Los resultados son de alguna manera interesantes, pero deben presentarse de forma temática o coherente, ya sea para responder a la pregunta de investigación o para probar la hipótesis. Sin embargo, se debe hacer una discusión extensa al final de la literatura.
Respuesta 5: Proporcione su respuesta para el Punto 5.
Las discusiones se actualizaron agregando la literatura que respalda los resultados de la investigación. Se adjunta el artículo actualizado.Certificado  

Reviewer 2 Report

Dear authors, I am sending comments to the articles:
COVID-19 and its Effects on the Management of Basic Quality Conditions
in the Universities of Peru, 2022 1. Please correct the abstract. Please write what period was examined

2.Please write more in the introduction about the impact of Covid
on education and various industries.
Please use more literature, e.g

Salehi, M.; Zimon, G.; Ghaderi, A.R.; Hasan, Z.A. The Relationship between Prevention and Panic from COVID-19, Ethical Principles, Life Expectancy, Anxiety, Depression and Stress. Int. J. Environ. Res. Public Health 2022, 19, 5841. https://doi.org/10.3390/ijerph19105841
3. Please also write something about sustainable development
in the introduction and the problems you describe
4.Please highlight the most important conclusions with discussions.
There is no discussion in the conclusions,
you have to improve it, it is very important.
These are very weak conclusions at the moment.

5. P
lease use more literature in the introduction and conclusion.

Sincerely
  ikona Zweryfikowane przez spoÅ‚eczność

Author Response

Response to Reviewer 2 Comments
(x) English language and style is ok/minor spell check required.
The MDPI Language Editing Services translator was used.
Point 1:
1. Please correct the abstract. Please write what period was examined
Answer 1:Provide your answer for Item 1.
The summary has been rewritten incorporating the importance of the topic, the period that was examined as well as the methodological approach. The article updating the summary is attached.
Point 2:
2.Please write more in the introduction about the impact of Covid on education and various industries.
Answer 2: Provide your answer for Item 2.
The suggested bibliography has been incorporated, as well as in the introduction on the impact of Covid.
in education and various industries. The article updating the translation is attached.
Point 3:
3. Please also write something about sustainable development in the introduction and the problems you describe
Answer 3:Provide your answer for Item 3.
The definitions of sustainable development have been incorporated, as suggested, a section of the problems that describe it has been incorporated. The article is attached, updating what is indicated in the introduction and in point 1.1.
point 4
4.Please highlight the most important conclusions with discussions.
There is no discussion in the conclusions,
you have to improve it, it is very important.
These are very weak conclusions at the moment.
Answer 4:Provide your answer for Item 4.
2
The most important conclusions have been highlighted with the discussions as indicated.
The updated article is attached
point 5
5. Please use more literature in the introduction and conclusion.
The literature was updated in the introduction and conclusions. The updated article is attached.CERTIFICATE

Reviewer 3 Report

Too long a theory, basic terms would like to be defined more specifically.

Is there perhaps some difference between education in the south of the country and the rest of the country - it is still emphasized in the south of the country.

Hypotheses seem too general, it keeps repeating itself in the south of the country

Author Response

Response to Reviewer 3 Comments
Point 1:
1. Too long a theory, basic terms would like to be defined more specifically.
Answer 1:Provide your answer for Item 1.
The suggestion of the point has been incorporated into the article. The updated article is attached.
Point 2:
2. Is there perhaps some difference between education in the south of the country and the rest of the country - it is still emphasized in the south of the country.
Answer 2: Provide your answer for Item 2.
It is explained in more detail in point 1.1. Research Problem, Objectives and Questionsof the article the observation given.The updated article is attached.
Point 3:
3. Hypotheses seem too general, it keeps repeating itself in the south of the country
Answer 3:Provide your answer for Item 3.
The hypotheses have been improved as indicated. The updated article is attached.CERTIFICATE

Reviewer 4 Report

Hi:

after reading this article, I confirm that it your work is well structured, professionally written and presents the best results on the problem of teaching in the university during covid-19. 

Juste please add this reference:

Disruption Caused by the COVID-19 Pandemic in Peruvian University Education. November 2020.International Journal of Higher Education 9(9):80. DOI: 10.5430/ijhe.v9n9p80.

Good Luck

Author Response

Response to Reviewer 4 Comments
After reading this article, I confirm that it your work is well structured, professionally written and presents the best results on the problem of teaching in the university during covid-19.
Juste please add this reference:
Point 1:
Answer 1: Provide your answer for Item 1.
The suggested bibliography has been incorporated,The updated article is attached.CERTICATE

Round 2

Reviewer 1 Report

Dear author, thanks for sharing the revised copy. 

Author Response

Dear reviewer,
Thank you for your comments on the revision of the document.
The updated papper is attached.
Yours respectfully,

Reviewer 2 Report

Dear Authors,

I have no more comments on the article

Sincerely

Author Response

Dear reviewer,
Thank you for your review comments, they have been very helpful.
The updated document is attached.
Respectfully yours,

Reviewer 3 Report

The language of the article has been edited.

The theoretical part was not shortened, but on the contrary expanded - it is perhaps more understandable, however, the idea of sustainability of universities could be better specified, at least in one sentence at the end.

Author Response

Point 1:
The theoretical part was not shortened, but on the contrary expanded - it is perhaps more understandable, however, the idea of sustainability of universities could be better specified, at least in one sentence at the end.
Answer 1: Provide your answer for Item 1.
Dear reviewer, we appreciate your suggestion, precisely the other three reviewers suggested depth in the development of the theoretical framework covering the topic of sustainability and universities.
A sentence has been incorporated that complements how our findings are within the framework of the sustainability of universities
